# LUT-GEMM: Quantized Matrix Multiplication based on LUTs for Efficient Inference in Large-Scale Generative Language Models

**Gunho Park[1*], Baeseong Park[2*], Minsub Kim[2], Sungjae Lee[2], Jeonghoon Kim[2],**
**Beomseok Kwon[2], Se Jung Kwon[2], Byeongwook Kim[2], Youngjoo Lee[1], Dongsoo Lee[2]**
[1] Pohang University of Science and Technology
[2] NAVER Cloud
{gunho1123,youngjoo.lee}@postech.ac.kr
{baeseong.park,minssub.kim,sung-jae.lee,jeonghoon.samuel,
beomseok.kwon,sejung.kwon,byeonguk.kim,dongsoo.lee}@navercorp.com

## Abstract

Recent advances in self-supervised learning and the Transformer architecture have significantly improved natural language processing (NLP), achieving remarkably low perplexity. However, the growing size of NLP models introduces a memory wall problem during the generation phase. To mitigate this issue, recent efforts have focused on quantizing model weights to sub-4-bit precision while preserving full precision for activations, resulting in practical speed-ups during inference on a single GPU. However, these improvements primarily stem from reduced memory movement, which necessitates a resource-intensive dequantization process rather than actual computational reduction. In this paper, we introduce LUT-GEMM, an efficient kernel for quantized matrix multiplication, which not only eliminates the resource-intensive dequantization process but also reduces computational costs compared to previous kernels for weight-only quantization. Furthermore, we proposed group-wise quantization to offer a flexible trade-off between compression ratio and accuracy. The impact of LUT-GEMM is facilitated by implementing high compression ratios through low-bit quantization and efficient LUT-based operations. We show experimentally that when applied to the OPT-175B model with 3-bit quantization, LUT-GEMM substantially accelerates token generation latency, achieving a remarkable $2.1\times$ improvement on a single GPU when compared to OPTQ, which relies on the costly dequantization process. The code is available at https://github.com/naver-aics/lut-gemm

## 1 Introduction

Recent years have observed large-scale language models (LLMs) presenting state-of-the-art performance on various natural language process (NLP) tasks. Such rapid progress in NLP performance has been highly facilitated by the self-supervised learning methods, avoiding expensive manual labeling (Devlin et al., 2019; Baevski et al., 2020; Chen et al., 2020). Leveraging extensive training datasets, these models benefit from efficient sequence-to-sequence architectures like the Transformer model (Vaswani et al., 2017), leading to notable increases in model parameters.

Previous studies (Brown et al., 2020; Kaplan et al., 2020; Hoffmann et al., 2022) have reported that LLM performance follows a predictable power-law scaling as a function of model size. Accordingly, in recent years, several large-scale generative language models, including GPT-3 (175B) (Brown et al., 2020), HyperCLOVA (204B) (Kim et al., 2021a), Gopher (280B) (Rae et al., 2021), Chinchilla (70B) (Hoffmann et al., 2022), Megatron Turing NLG (530B) (Smith et al., 2022), PaLM (540B) (Chowdhery et al., 2022), and LLaMA (65B) (Touvron et al., 2023), have been proposed to further advance state-of-the-art performance. However, models with billions of parameters cannot be accommodated on a single GPU due to the limited memory size of GPUs, which is sacrificed to

---

*Equal contribution

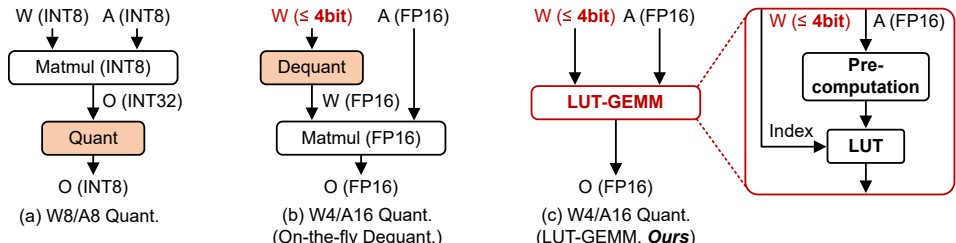

Figure 1: Three matrix multiplication schemes for W8/A8 or W4/A16 (i.e., weight-only) quantization format. Our proposed LUT-GEMM can adopt W4 and A16 without requiring an additional dequantization step.

enhance memory bandwidth (Migacz, 2017; Yu et al., 2017). To address such a concern, researchers have proposed to use model parallelism, which distributes computations over multiple GPUs through GPU-to-GPU communication (Shoeybi et al., 2019; Narayanan et al., 2021). Nevertheless, it is worth noting that model parallelism introduces additional overheads, stemming from the inter-GPU communication. Consequently, the performance gains achieved through model parallelism exhibit a sub-linear relationship with the number of GPUs employed.

To mitigate the challenges related to model parallelism, parameter quantization (Xu et al., 2018; Choi et al., 2017; McDonnell, 2018) presents a practical solution for minimizing model size, reducing the number of GPUs required for inference. Among the various quantization schemes, the preferred choice is quantizing both activations and weights to exploit integer-based arithmetic units (Jacob et al., 2018; Wu et al., 2018; Lin et al., 2016). Nonetheless, this quantization method is practically limited to 8 bits, and non-linear operations (e.g., softmax and normalization) may yield imprecise results (Kim et al., 2021b; Bhandare et al., 2019). Moreover, to fully utilize integer arithmetic units, it is essential to implement on-the-fly activation quantization and dequantization, along with an accurate estimation of the activation distribution (Yao et al., 2022; Dettmers et al., 2022).

Recent research has proposed 4-bit weight-only quantization as an approach for memory compression (Frantar et al., 2022; Lin et al., 2023; Dettmers et al., 2023; Kim et al., 2023), involving on-the-fly conversion to full-precision. While this sacrifices the computational benefits of using integer arithmetic, empirical findings in LLMs suggest that weight-only quantization can achieve significantly higher compression ratios for a given target accuracy compared to quantizing both weights and activations (Zeng et al., 2022). Various weight-only quantization methods have been proposed to improve the compression ratio while preserving accuracy, often accompanied by dedicated kernels for practical acceleration through quantization (Jeon et al., 2020; Frantar et al., 2022; Lin et al., 2023).

In this paper, we present LUT-GEMM, a kernel designed to facilitate quantized matrix multiplications with quantized weights and full-precision activations. As shown in Figure 1, LUT-GEMM addresses two issues prevalent in previous quantization approaches: 1) accuracy degradation due to quantized activations and 2) the need for additional dequantization implementation. LUT-GEMM inherently accommodates quantized weights and full-precision activations, enabling the acceleration of the inference process while preserving the desired level of precision. Specifically, LUT-GEMM employs the binary-coding quantization (BCQ) format (Rastegari et al., 2016) to capitalize on simple arithmetic operations. It is worth noting that BCQ was initially proposed to support non-uniform quantization, which relies on customized hardware for bit-level operations. To our knowledge, we are the first to show that prior uniform quantization can be reformulated in the form of BCQ, allowing LUT-GEMM to support both non-uniform and uniform quantization formats. Consequently, LUT-GEMM can execute a wide range of weight-only quantization schemes for matrix multiplications, achieving low inference latency and eliminating the need for on-the-fly dequantization.

Our major contributions in this work include the following: **1)** We verify that BCQ is capable of representing both uniform and non-uniform weight quantization. **2)** We show that LUT-GEMM, using the BCQ format, offers a broad spectrum of latency and accuracy trade-offs, leveraging GPU-specific hardware utilization methods to implement various BCQ configurations efficiently. **3)** For LLMs, we demonstrate that LUT-GEMM, which utilizes quantized weights *without a dequantization process*, can considerably accelerate matrix multiplications with small quantization bits while power consumption is greatly saved by reducing the number of GPUs. **4)** Assuming a 3-bit BCQ format for weights of OPT-175B served by a single GPU, experimental results show that LUT-GEMM accelerates token generation latency by $2.1\times$ compared to the OPTQ (Frantar et al., 2022).

## 2 BACKGROUND

### 2.1 GPU-ACCELERATED GENERATIVE LMS

For large LMs, the processing time of matrix multiplications dominates the entire inference latency because of higher time complexity compared to activation functions, normalization layers, and so on (Dettmers et al., 2022). Specifically, matrix multiplications account for at least 75% of the processing time for various LM sizes and input token lengths (see Appendix A). Note that due to the limited memory capacity of a single GPU, large LMs may need multiple GPUs, resulting in increased communication latency. GPUs are commonly adopted to accelerate inference as GPUs embed lots of arithmetic units and support multiple threads, critical for speeding up matrix multiplications (Narayanan et al., 2021; Migacz, 2017). However, extracting high performance from GPUs depends on arithmetic intensity, and therefore, the batch size should be large enough to ensure a high reuse ratio from main memory (Markidis et al., 2018).

### 2.2 QUANTIZATION METHODS AND LIMITATIONS

Various research efforts have been made to enhance the serviceability of large and heavy deep neural networks by improving their latency and throughput. These efforts include quantization (Rastegari et al., 2016; Jacob et al., 2018; Nagel et al., 2017; Xu et al., 2018; Chung et al., 2020), pruning (Han et al., 2016; Zhu & Gupta, 2017; Gale et al., 2019), knowledge distillation (Hinton et al., 2015; Polino et al., 2018), and low-rank approximation (N. Sainath et al., 2013; Chen et al., 2018; Edalati et al., 2021). Among these, quantization is the most extensively researched field, which involves using faster and more efficient computing units and reducing memory usage. Uniform quantization using an INT8 operator is particularly well-studied among the various quantization formats and is currently being actively applied in LLMs (Yao et al., 2022; Dettmers et al., 2022; Xiao et al., 2022).

INT8 arithmetic units, commonly found in contemporary computing systems, offer reduced latency (thanks to their low hardware complexity) and decreased memory usage of up to 50% compared to FP16. Thus, present NVIDIA GPUs employ Tensor cores that support INT8 operations to accelerate computations (Markidis et al., 2018). However, the utilization of INT8 mandates the quantization of activations, which can pose a significant challenge in achieving compressed and accelerated models employing INT8 units. For instance, when scaling factors are determined offline, activation quantization may lead to a considerable reduction in model accuracy as outliers are approximated.

To address declining accuracy in quantizing LLMs, token-based dynamic quantization has emerged as a crucial technique (Yao et al., 2022; Dettmers et al., 2022). The LLM.int8() method (Dettmers et al., 2022) addresses the systematic appearance of outliers in LLMs by proposing a decomposition method that conducts most computations in INT8 and dequantizes a limited number of outliers to FP16, resulting in only a marginal latency decrease in LLMs. SmoothQuant (Xiao et al., 2022) takes an advanced approach by mathematically transferring activation variance (challenging to quantize due to outliers) to weights, which are relatively easier to quantize. This technique enables efficient computation using INT8 arithmetic units (i.e., W8A8), even when employing static quantization of activations. However, it should be noted that applying SmoothQuant in novel LLMs is necessary to empirically observe and validate the occurrence of outliers before applying the migration technique.

The utilization of INT8 precision introduces variability in its efficacy, primarily influenced by the specific characteristics of each phase within LLM inference. While the summarization phase exhibits a high likelihood of weight reuse, resulting in a compute-bound operation, the generation phase, due to its autoregressive nature, primarily involves a memory-bound workload. This intrinsic memory limitation restricts hardware throughput even with INT8 precision. Consequently, services generating a large number of tokens may experience only marginal performance gains with INT8 adoption.

Recent studies have focused on the inefficiency of the generation step and, in response, proposed the utilization of the W4A16 format (Frantar et al., 2022; Zeng et al., 2022; Dettmers et al., 2023; Kim et al., 2023), which compresses model weights into 4-bit integers without quantizing the activations as weights typically dominate memory size compared to activations. Given that current computing systems cannot accelerate the W4A16 format, OPTQ (Frantar et al., 2022) and AWQ (Lin et al., 2023) solve such an issue by dynamically dequantizing weights to FP16 and then performing matrix multiplication. Leveraging the memory-bound nature of the generation step, this approach enhances generation latency by reducing memory movement despite the dequantization overheads.

### 2.3 BINARY-CODING QUANTIZATION

Binary-coding quantization (BCQ) initially introduced by Xu et al. (2018), presents a compelling alternative to conventional uniform quantization methods. For instance, when a weight vector $\boldsymbol{w}$ (of size $n$) is quantized into $q$-bit by BCQ, $\boldsymbol{w}$ is approximated to be $\sum_{i=1}^{q} \alpha_i \boldsymbol{b}_i$ where $\alpha_i \in \mathbb{R}^+$ is a scaling factor and $\boldsymbol{b}_i \in \{-1, +1\}^n$ is a binary vector. In this paper, we have chosen to employ BCQ as our primary quantization technique for weights while retaining activations in full precision to address the challenges mentioned earlier. Moreover, we introduce an extension to BCQ, enhancing its capabilities to encompass both uniform quantization and group-wise quantization. This extension, outlined in the subsequent section, not only broadens BCQ's utility but also enables the applicability of LUT-GEMM to various quantization methods.

## 3 DESIGN METHODOLOGY OF LUT-GEMM

LUT-GEMM is devised to develop high-performance, energy-efficient inference systems for LLMs. To achieve this objective, LUT-GEMM incorporates several innovative approaches. Firstly, we employ a lookup table (LUT) based computation technique to mitigate redundant calculations caused by digitized binary weights after BCQ. Furthermore, since most non-uniform quantization methods involve complex operations with limited parallelism and often lack hardware support, we design LUT-GEMM to efficiently support BCQ formats. Our proposed kernel, LUT-GEMM, directly utilizes BCQ formats without additional overhead, such as dequantization. Secondly, we expand conventional BCQ methods by introducing a bias term, significantly enhancing representational capability. This simple yet profound enhancement enables the representation of both non-uniform and uniform quantization methods within the extended BCQ format, providing us with the flexibility to leverage various quantization techniques based on the specific requirements of LLMs. Finally, We further refine the implementation details of the binary-coding quantization scheme, enabling a trade-off between compression ratio and quantization error to better exploit the characteristics of LLMs. As a result, LUT-GEMM demonstrates reduced latency and/or a decreased number of GPUs required for LLM inference while inherently accommodating various weight-only quantization methods.

### 3.1 LUT BASED QUANTIZED MATRIX MULTIPLICATION

Our quantization scheme, which utilizes BCQ format for weight-only quantization while maintaining full precision for activations, leads to duplicate and redundant partial computations in naive matrix multiplications. To illustrate, assume that a binary matrix $\boldsymbol{B} \in \{-1, +1\}^{4 \times 6}$ and an activation vector $\boldsymbol{x} \in \mathbb{R}^6$ are given as

$$\boldsymbol{B} = \begin{bmatrix} +1 & +1 & -1 & -1 & -1 & +1 \\ +1 & +1 & -1 & +1 & +1 & -1 \\ +1 & +1 & -1 & -1 & -1 & -1 \\ -1 & -1 & +1 & -1 & -1 & +1 \end{bmatrix}, \quad \boldsymbol{x} = \begin{bmatrix} x_1 & x_2 & x_3 & x_4 & x_5 & x_6 \end{bmatrix}. \quad (1)$$

Then, computing $\boldsymbol{B}\boldsymbol{x}^\top$ (that is to be multiplied by scaling factors) would repeat $(x_1 + x_2 - x_3)$ three times and $(-x_4 - x_5 + x_6)$ two times. Such redundant computations are caused by digitized elements of $\boldsymbol{B}$, and thus, we expect more duplicated computations as the size of matrices increases according to the growth of model size. Moreover, loading each element of $\boldsymbol{B}$ requires bit-level memory accesses that can be slow for commercial CPUs and GPUs.

To efficiently perform $\boldsymbol{B}\boldsymbol{x}^\top$ while avoiding bit-level memory accesses, we can pre-compute all possible combinations of full-precision activations and binary patterns. Note that a LUT has been widely used to save processing time when numerous computations yield outputs within a restricted set (de Queiroz & Stein, 2004; Meher, 2010; Jeon et al., 2020; Xu et al., 2021). LUT-based computation is justified, especially when retrieving a value from a LUT is much faster than carrying out the original calculations. BCQ format (without quantizing activations that require heavy modifications in training codes and model structure (Wu et al., 2018; Jacob et al., 2018)) is also useful to be implemented by LUT-based approaches. To construct a LUT, a hyperparameter $\mu$ is introduced, representing the sub-vector length of $\boldsymbol{x}$. For example, to address redundant computation observed in Equation 1, we can pre-compute 8 (=$2^3$) possible values with every three elements in $\boldsymbol{x}$ and store those values in a LUT. Hence, $\mu$ is 3 in this example. Once $2^\mu$ values of a LUT are generated by

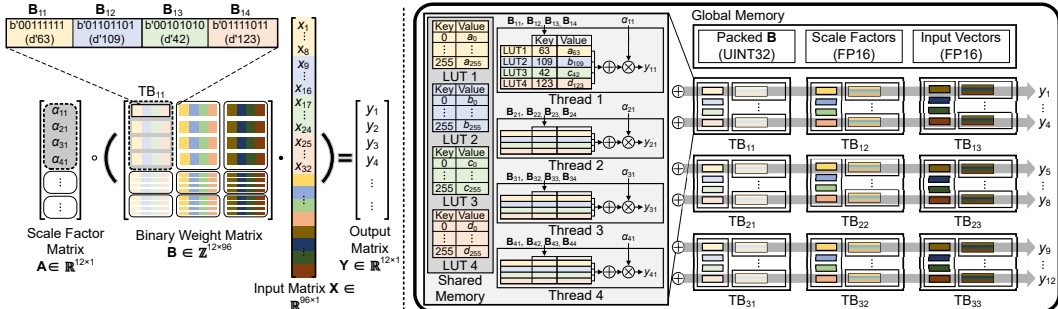

Figure 2: The overview of LUT-GEMM implementation on GPUs. In this example, we assume $\mu = 8$, $t_h = 4$, $l = 4$, $t_w = 32$, and $q = 1$. "∘" denotes element-wise multiplication and "·" indicates a tensor product.

using a sub-vector of $x$, arithmetic operations to obtain partial dot products (of $Bx^\top$) are replaced with LUT retrieval operations while a key is given by concatenating $\mu$ binary elements of $B$. To complete $Bx^\top$ computation, as the final step, those partial products are summed and then multiplied by scaling factors. When the row dimension of $B$ is enlarged (as generative LMs get larger), the utilization of a LUT increases due to more occurrences of redundant computations.

To quantify the reduction in computation, assume that a $q$-bit quantized $(m \times n)$ binary matrix $B_i$ (for $i = 1, 2, 3, \ldots, q$) is multiplied with an $(n \times 1)$ input vector $x$ using $\mu$-width LUTs. The computational complexity of the LUT-based approach can be described as follows, when $mq \gg 2^\mu$:

$$C = C_{build} + C_{read} = \mathcal{O}\left(2^\mu \cdot \frac{n}{\mu} + m \cdot \frac{n}{\mu} \cdot q\right) \approx \mathcal{O}\left(m \cdot \frac{n}{\mu} \cdot q\right), \tag{2}$$

where $C_{build}$ and $C_{read}$ represent the complexity of building the LUT and reading a value from the LUT, respectively. Consequently, compared to the complexity of conventional matrix multiplication $\mathcal{O}(mn)$, LUT-GEMM can achieve a computational savings of $\frac{q}{\mu}$ times by leveraging LUTs.

## 3.2 LUT Based implementation on GPU

In addition to the LUT-based scheme (eliminating redundant computations and bit-level memory accesses), our strategy for optimizing single-batch operations on GPUs is as follows: 1) To improve parallelism, we create as many threads as possible while each thread is allowed to perform independent LUT accesses. 2) Binary weights accessed by a thread can share a common scaling factor such that operations related to scaling factors do not degrade the performance of a thread. 3) If we allocate too small resources to a thread, then LUT utilization can be low, and synchronization overhead can increase. As such, we need to optimize thread configurations empirically.

Figure 2 illustrates the overall LUT-GEMM implementation scheme on GPUs. For the sake of simplicity, we formulate the proposed quantized matrix multiplication as $\mathbf{y} = \sum_{i=1}^{q} (A_i \circ (B_i \cdot x))$, where $A$ is an $(m \times 1)$ FP16 scaling matrix, $B$ is an $(m \times n)$ FP16 binary matrix, $x$ is an FP16 input vector of size $n$, and the operator $\circ$ indicates element-wise multiplication. For LUT-GEMM, we assign $l$ LUTs to a thread block (TB) of GPU. Then, the size of submatrix of $B$ allocated to each TB becomes $(t_h \times t_w)$ where $t_w = l \times \mu$. Small $t_h$ can increase the number of available threads while large $t_h$ enhances LUT utilization inside a TB. Thus, $t_h$ is empirically determined (2048 is a practical number for large-scale LMs). For $q > 1$, the entire process of Figure 2 can be iterated $q$ times while intermediate results are accumulated. See Appendix B for implementation details.

## 3.3 Representational Capability of LUT-GEMM

LUT-GEMM extends the BCQ format to support both non-uniform and uniform quantization methods, which are widely used in compressing LLM. In this section, we introduce extended BCQ format to enhance representational capability of LUT-GEMM.

**Asymmetric Binary-Coding Quantization** Conventional BCQ methods are basically limited to exhibiting symmetric quantization shape with respect to the zero point. Another constraint of BCQ

methods stems from their inability to represent the value of zero due to the symmetric quantization properties. To enhance the representational capability of BCQ, we extend the conventional BCQ method by including a bias term $z$ as follows:

$$\hat{\boldsymbol{w}} = \sum_{i=0}^{q-1} (\alpha_i \cdot \boldsymbol{b}_i) + z, \ \boldsymbol{b}_i \in [-1, +1]^n, \tag{3}$$

where a weight vector (of size $n$) is decomposed into binary vector $\boldsymbol{b}_i$ with associated scaling factors $\alpha_i$. As a result, the modified BCQ format, along with the inclusion of the bias term, can represent asymmetry quantization centered around z, as illustrated in Figure 3(a).

**Uniform Quantization**   Incorporating a bias term into the BCQ format, we have discovered that the extended BCQ format can effectively represent uniform quantization by carefully adjusting the scaling factors and bias term, as visually depicted in Figure 3(b). See Appendix C for details of how asymmetric uniform quantization can be converted into symmetric BCQ with a bias term. In Figure 3, we can see that for $q$-bit quantization, the uniform quantization method employs a single scaling factor, while the non-uniform quantization technique calls for the use of $q$ distinct scaling factors. Consequently, due to the incorporation of the extended binary-coding quantization format, our proposed LUT-based matrix multiplication scheme is applicable to both uniform quantization and binary-coding-based non-uniform quantization methods.

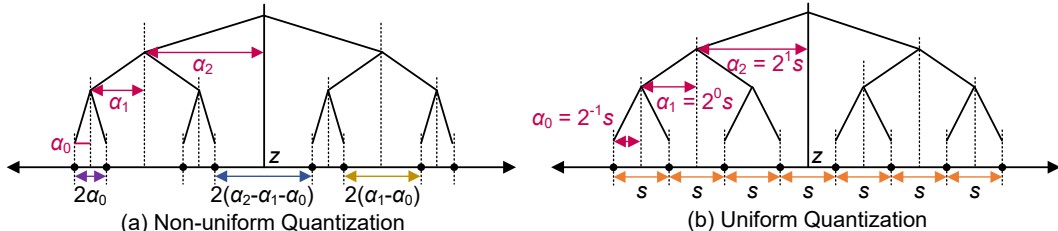

Figure 3: Extension of binary-coding quantization to support both non-uniform and uniform quantization formats, achieved by including a bias term ($q = 3$).

### 3.4   Latency-Accuracy Trade-Off for Improved Applicability

As the hidden size increases rapidly (e.g., $d_{\text{model}} = 12288$ for GPT-3 175B) according to the advent of large-scale LMs, it would be more difficult to compute a proper scaling factor shared by a larger number of weights. As an alternative to row-wise quantization, group-wise quantization in which a scaling factor can be shared by an arbitrary number of weights is widely used to minimize quantization error (Frantar et al., 2022; Lin et al., 2023). To examine the latency variance of LUT-GEMM with respect to group size $g$, we perform matrix multiplications (using an $(m \times n)$ matrix and an $(n \times 1)$ matrix) when $g$ values vary as shown in Figure 4(a). We observe a sufficiently large $g$, such as 128 and 256, can result in fast LUT-GEMM while accuracy improvement by group-wise is substantial. To gain insights into the underlying mechanisms, we analyze the memory footprint of LUT-GEMM because single-batch operations are primarily memory-bound and latency is proportional to the memory footprint. Let $S_b$ and $S_\alpha$ represent the space complexity of binary weights and scaling factors, respectively. Then the overall space complexity $S$ can be described as

$$S = S_b + S_\alpha = \mathcal{O}\left(1 \cdot m \cdot n \cdot q + 16 \cdot m \cdot \frac{n}{g} \cdot q\right) = \mathcal{O}\left(m \cdot n \cdot q \left(1 + \frac{16}{g}\right)\right). \tag{4}$$

As a consequence, when $g \gg 16$, $S$ becomes independent of $g$ and can be approximated as $\mathcal{O}(m \cdot n \cdot q)$. To verify our claim that latency of LUT-GEMM is proportional to memory footprint (when running single-batch operations), we explore various $(q, g)$ pairs and their corresponding compression ratios and measure matrix multiplication latency when $m = n = 12288$ as depicted in Figure 4(b). Note that even if the number of quantization bits $q$ is smaller, the memory footprint can be large when the group size $g$ is small (i.e., more scaling factors are employed, see Appendix D). Therefore, the additional search parameter $g$ allows a fine-grained search space of compression ratio that is not available by $q$ alone. Across all available compression ratios, latency is a function of compression ratio. For instance, if two different pairs $(q_1, g_1)$ and $(q_2, g_2)$ exhibit a similar memory footprint, then we can expect similar latency by LUT-GEMM.

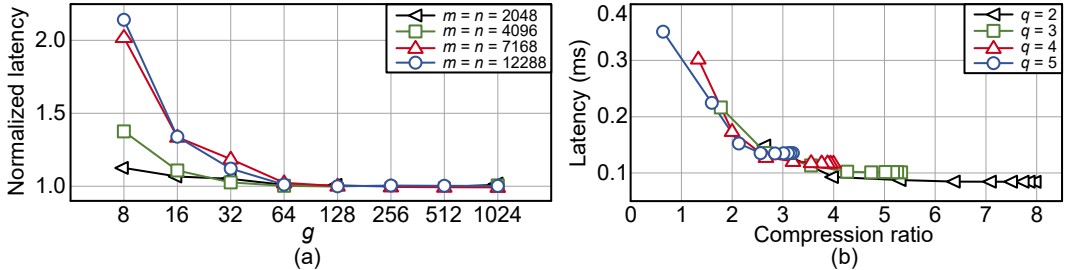

Figure 4: (a) Normalized LUT-GEMM latency when a $(m \times n)$ weight matrix is quantized by 3-bit with different $g$ values. (b) Relationship between latency and compression ratio when LUT-GEMM performs quantized matrix multiplications with $m = n = 12288$ and various $(q, g)$ pairs.

## 4 EXPERIMENTAL RESULTS

In this section, we present the experimental results obtained by utilizing LUT-GEMM across various levels of complexity, ranging from a single-layer experimental setup to the complete model level. Initially, we examine the influence of LUT-GEMM on a specific layer, followed by an investigation into the inefficiency of tensor parallelism. Finally, we analyze the end-to-end latency of OPT models (Zhang et al., 2022) to determine the overall effect of LUT-GEMM on performance.

### 4.1 KERNEL EVALUATION

**Latency Comparisons with Various Kernels**  Table 1 shows latency measurements for the first layer of the Feed-Forward Network (FFN) in the OPT-175B model (Zhang et al., 2022). The measured kernels include cuBLAS (for FP-FP or INT-INT), OPTQ (Frantar et al., 2022), AWQ (Lin et al., 2023) (for FP-INT), and LUT-GEMM (for FP-INT or FP-BCQ). Note that OPTQ and AWQ kernels involve dequantization followed by GEMM, while LUT-GEMM accepts quantized weights directly, eliminating the need for dequantization. We can observe that the latency of the INT8-INT8 (with cuBLAS) implementation only slightly improves latency over FP-FP since cuBLAS is not well-optimized for single batch operations. While OPTQ and AWQ achieve lower latency than cuBLAS due to reduced memory movement but are slower than LUT-GEMM due to dequantization overhead. Moreover, LUT-GEMM exhibits $2.6\times$ reduction in computation compared to the previous GEMM kernels. Thus, the proposed LUT-GEMM kernel, which operates directly on quantized weights and reduces computational complexity, achieves the lowest latency among the kernels considered.

Table 1: Latency comparison of the first FFN layer on OPT-175B model with various precision and corresponding kernel selections with $m = 12288$ and $g = 128$ on A100-80GB-GPU.

| Kernel | Precision for Data Type | | | Latency (ms) (Speed up) |
|---|---|---|---|---|
| | Weight $(4m \times m)$ | Input $(m \times 1)$ | Output $(4m \times 1)$ | |
| cuBLAS | FP32 | FP32 | FP32 | 1.4015 ($\times 0.52$) |
| cuBLAS | FP16 | FP16 | FP16 | 0.7256 ($\times 1.00$) |
| cuBLAS | INT8 | INT8 | INT32 | 0.6345 ($\times 1.14$) |
| LUT-GEMM | INT8* | FP16 | FP16 | 0.4620 ($\times 1.57$) |
| OPTQ (Frantar et al., 2022) | INT3 | FP16 | FP16 | 0.3599 ($\times 2.02$) |
| AWQ (Lin et al., 2023) | INT4 | FP16 | FP16 | 0.3238 ($\times 2.24$) |
| LUT-GEMM | INT4* | FP16 | FP16 | 0.2688 ($\times 2.70$) |
| LUT-GEMM | INT3* | FP16 | FP16 | 0.2250 ($\times 3.22$) |

[*] LUT-GEMM supports both non-uniform and uniform quantization.

**Comparison with FP16 Tensor Parallelism**  Table 2 summarizes the profiling results of matrix multiplications performed by using cuBLAS (with tensor parallelism) or LUT-GEMM. GPU power and other metrics are collected by using *nvidia-smi* utility (Tiwari et al., 2015; Ali et al., 2020). We notice that tensor parallelism with additional GPUs in cuBLAS brings about a significant decrease in GPU utilization, memory bandwidth (BW) utilization, and computation latency ratios. As evidenced by the increase in the latency ratio of communication, such reductions in utilization indicate that

some GPUs can be temporarily idle until all GPUs are synchronized. Accordingly, the speed-up that can be obtained by tensor parallelism is a lot smaller than the number of GPUs. As a result, cuBLAS with more GPUs causes increased energy consumption for matrix multiplications. On the other hand, LUT-GEMM (with one GPU) can offer high speed-up (that cannot be achieved by tensor parallelism) while retaining high GPU/memory utilization. Combining low latency and a reduced number of GPUs, thus, LUT-GEMM substantially saves energy consumption for matrix multiplications.

Table 2: Profiling results of matrix multiplications (with an $(4m \times m)$ matrix and an $(m \times 1)$ matrix). For LUT-GEMM, $g = m$ and $q$ is 2 or 4.

| Kernel-GPUs-$m$-$q$ | Comm. Ratio (%) | Speed Up | GPU BW Util. (%) | Memory Util. (%) | Avg. Power (W/GPU) | Total Energy (mJ) |
|---|---|---|---|---|---|---|
| cuBLAS-1-12288-16 | 0.00 | 1.00 | 96.57 | 98.37 | 215.70 | 161.85 ($\times$1.00) |
| cuBLAS-8-12288-16 | 38.58 | 3.45 | 52.48 | 24.27 | 172.44 | 299.96 ($\times$1.85) |
| LUT-GEMM-1-12288-2 | 0.00 | **4.85** | 88.63 | 58.38 | 280.85 | 43.49 ($\times$**0.27**) |
| LUT-GEMM-1-12288-4 | 0.00 | **3.23** | 92.51 | 77.98 | 292.76 | 68.09 ($\times$**0.42**) |

## 4.2 END-TO-END LATENCY

We now evaluate the end-to-end latency of inference with a single batch size, considering various LLaMA models with quantized weights while preserving full precision activations. Table 3 illustrates the end-to-end latency per token and perplexity when weights are uniformly quantized using AWQ method (Lin et al., 2023) . Additionally, the evaluation for OPT family models and LLaMA family models can be found in Appendix Table 6 and Table 7, respectively, providing a comprehensive overview. Latency measurements are conducted within the FasterTransformer framework, exploring different GPU configurations to assess potential speed-up gains from model parallelism. From our observations, we can conclude the following: 1) Reducing the group size ($g$) effectively decreases perplexity, even when employing a simple RTN quantization scheme, at the cost of a marginal increase in latency, 2) Increasing the number of GPUs (and, consequently, parallelism) does not significantly reduce latency due to various overheads such as GPU-to-GPU communication cost, as described in Appendix A. In the case of LUT-GEMM, where matrix multiplications are accelerated, relative GPU-to-GPU communication overheads become more pronounced compared to cuBLAS, making model parallelism less effective. This highlights the prominence of communication overheads for high-performance matrix multiplication engines.

In the case of the LLaMA-30B model, end-to-end inference with FP16 weights can be executed on a single GPU (See Appendix Figure 8 for a latency comparison of various model sizes on a single GPU). However, for the LLaMA-65B model with FP16 weights, the model size exceeds the memory capacity of a single GPU (80GB for A100), necessitating model parallelism techniques. Nevertheless, when the weights of the LLaMA-65B model are quantized to 3 or 4 bits, as demonstrated to be a viable solution in (Frantar et al., 2022), inference can be accommodated on a single GPU. Assuming 3-bit quantization and the implementation of LUT-GEMM with $g$=128, a speed-up of 2.41$\times$ for LLaMA-30B (using one GPU) and 2.04$\times$ for LLaMA-66B (using two GPUs) is achievable. Note that when fine-tuning is performed after constructing a pre-trained model, more efficient quantization techniques such as AlphaTuning (Kwon et al., 2022) can enable even 2-bit quantization, as shown in Appendix Table 8. The efficacy of LUT-GEMM is further enhanced when combined with advanced quantization methods to reduce the number of quantization bits.

Table 3: Comparison of perplexity and end-to-end latency per token with cuBLAS and LUT-GEMM implementations with AWQ quantization method and model parallelism on multiple GPUs.

| Model | Kernel-$q$-$g$ | Quant. method | Perplexity Wiki2 | Latency (ms) 1-GPU | 2-GPU | 4-GPU |
|---|---|---|---|---|---|---|
| LLaMA 30B | cuBLAS-16-N/A | FP16 | 4.10 | 43.6 | 25.1 | 16.9 |
| | LUT-GEMM-4-128 | AWQ | 4.23 | 20.7 | 15.5 | 12.2 |
| | LUT-GEMM-3-128 | AWQ | 4.88 | 18.1 | 13.7 | 11.2 |
| LLaMA 65B | cuBLAS-16-N/A | FP16 | 3.53 | **OOM** | 46.9 | 29.1 |
| | LUT-GEMM-4-128 | AWQ | 3.67 | 35.7 | 25.4 | 19.9 |
| | LUT-GEMM-3-128 | AWQ | 4.24 | 31.3 | 23.0 | 18.3 |

## 5 Accelerating Quantized OPT-175B

Table 4 provides a comparison of the end-to-end latency for generating a token in OPT-175B, a representative large-scale LM, using the FasterTransformer framework. LUT-GEMM demonstrates its ability to decrease the number of GPUs needed for running inference, while concurrently reducing latency as $q$ diminishes or as the number of GPUs rises. For OPT-175B with FP16, a minimum of 8 GPUs is necessary for executing inference. However, upon utilizing the BCQ format for quantization, LUT-GEMM is able to perform inference using just a single GPU, while maintaining a comparable overall latency. It should be noted that, when comparing identical 3-bit (weight-only and row-wise) quantization scenarios, the latency for token generation using LUT-GEMM is $2.1\times$ lower than that of the OPTQ library. This significant reduction in latency can be primarily attributed to LUT-GEMM's ability to directly accept quantized weights, thereby eliminating the need for dequantization.

Table 4: End-to-end latency per token for OPT-175B model. The latency is measured on A100 80GB.

|  | | | Latency per token (ms) | | | |
| GPUs | Baseline | OPTQ | | LUT-GEMM | | |
| | FP16 | 3-bit | 1-bit | 2-bit | 3-bit | 4-bit |
| --- | --- | --- | --- | --- | --- | --- |
| 1 | **OOM** | 106.5 | 30.4 | 40.1 | 51.6 | **OOM** |
| 2 | **OOM** | N/A | 25.2 | 30.1 | 35.8 | 41.2 |
| 4 | **OOM** | N/A | 20.3 | 23.8 | 27.2 | 30.1 |
| 8 | 42.4 | N/A | 20.1 | 22.4 | 24.2 | 25.8 |

Let us demonstrate that the flexible features of LUT-GEMM (attributable to the extended BCQ format) can accelerate existing uniform quantization methods. Table 5 shows the perplexity of OPT-175B for various $q$ and $g$ configurations, as obtained using the OPTQ method (See Appendix Table 9 for a comprehensive accuracy evaluation at $q = 3$). The table also displays the corresponding latency (per generated token) achieved by LUT-GEMM (excluding FP16). The results clearly indicate that LUT-GEMM provides lower latency as $q$ decreases, although an excessively small $g$ may have a marginal adverse impact on latency. All in all, by integrating LUT-GEMM and OPTQ at the expense of an acceptable increase in perplexity, it is possible to reduce the number of GPUs required for running OPT-175B inference from eight GPUs to a single GPU while ensuring comparable latency. It is crucial to note that such an improvement in performance cannot be achieved when dequantization is included.

Table 5: Perplexity of quantized OPT-175B using OPTQ and end-to-end latency per token by LUT-GEMM for various $q$ and $g$ configurations. $g$ of '-' indicates row-wise quantization.

| Kernel | $q$ | $g$ | PPL[*] | Model size(GB) (Comp. Ratio) | Latency(ms) |
| --- | --- | --- | --- | --- | --- |
| cuBLAS | 16 | N/A | 8.34 | 347.9 ($\times$1.00) | 42.4 (8-GPU) |
| LUT-GEMM | 4 | - | 8.37 | 87.0 ($\times$**4.00**) | **OOM (1-GPU)** |
| | 3 | - | 8.68 | 65.3 ($\times$**5.33**) | 51.6 (**1-GPU**) |
| | 2 | 32 | 8.94 | 54.4 ($\times$**6.40**) | 55.2 (**1-GPU**) |
| | 2 | 64 | 9.18 | 48.9 ($\times$**7.11**) | 47.5 (**1-GPU**) |
| | 2 | 128 | 9.58 | 46.2 ($\times$**7.53**) | 46.5 (**1-GPU**) |

[*] PPL numbers are extracted from the OPTQ reference (Frantar et al., 2022).

## 6 Summary and Limitations

In this paper, we introduce LUT-GEMM, a highly efficient matrix multiplication kernel designed to operate directly on quantized weights, thereby eliminating the need for an additional dequantization step. Leveraging an extended BCQ format, LUT-GEMM exhibits the capability to process both uniformly and non-uniformly quantized models, achieving $2.1\times$ speedup over GPTQ. However, it is important to note some limitations: Our method primarily focuses on single-batch inference and exhibits diminishing performance gains as the batch size increases. This limitation is primarily attributed to the constrained memory bandwidth between core and LUTs in the shared memory. Nevertheless, we anticipate that this constraint will gradually diminish in significance with the advent of advanced hardware solutions.

## 7    REPRODUCIBILITY STATEMENT

In the Supplementary Materials, we furnish code for replicating our "Kernel Evaluation" experiments. Specifically, this entails:

- Offering our CUDA kernel alongside a concise benchmarking script for single matrix-vector products.
- Supplying a README document that presents example commands and instructions for executing all the scripts.

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

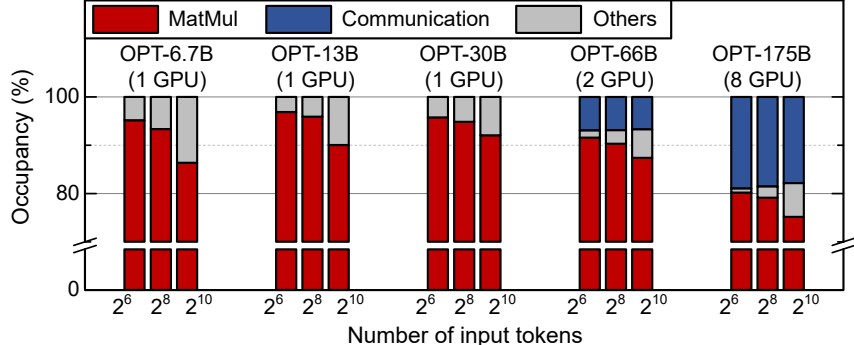

Figure 5: Latency breakdown of various OPT models. The overall performance is dominated by MatMul and communication operations. Experiments are conducted with A100 80GB and `FasterTransformer` (v4.0) framework.

## A   LLM INFERENCE LATENCY BREAKDOWN

Figure 5 presents a detailed breakdown of latency for various large-scale generative language models (LMs), specifically OPT models ranging from 6.7B to 175B parameters. This analysis considers different numbers of input tokens and was conducted on an A100 (80GB) GPU using the `FasterTransformer` inference framework developed by Nvidia[1]. The OPT models are built upon the Transformer architecture Vaswani et al. (2017) and consist of identical layers. Each layer incorporates multi-head attention and a feed-forward network, both involving four primary linear computations with relatively higher time complexity compared to other non-linear operations. Consequently, matrix multiplications constitute a substantial portion, accounting for at least 75% of the processing time across various LM sizes and input token lengths. Note that we can observe the emergence of GPU communication overhead in larger models. This is primarily attributable to the limited memory capacity of a single GPU, which necessitates communication between GPUs to accommodate the larger models.

## B   DETAILED IMPLEMENTATION

Each TB first conducts pre-computation using partial $x$ values assigned in order to fill up the $l$ number of LUTs. Then $l$ LUTs can be shared by all threads inside a TB (so as to mitigate costly global memory accesses) and multiple rows of a submatrix of $B$ can be processed by multiple threads (so as to improve throughput). When threads finish retrieving and summing LUT values, scaling factors are fetched (only once for each thread) and multiplied to produce partial outputs. Finally, $\frac{n}{l \times \mu}$ partial outputs are accumulated across TBs (through `atomicAdd` operations, as illustrated in Figure 2) to generate the final outputs. LUTs are stored in shared memory inside GPU and the shared memory presents high bandwidth (e.g., 19TB/s for A100). Thus, high memory accesses for LUTs (while multiple FLOPs can be replaced with one LUT access) enable fast matrix computations. As for the memory size of LUTs, only 1KB is required for every 8 hidden dimensions and the shared memory size is more than a few megabytes (e.g., 20MB for A100 with 192KB per SM and 108 SMs available). Thus, the whole LUTs can be safely stored in shared memory. To illustrate, the hidden dimension can be up to 324,000 for A100 while 12,288 is the hidden dimension for GPT-3 175B.

## C   CONVERSION OF UNIFORM QUANTIZATION INTO BCQ

A $q$-bit uniformly quantized weight $\hat{w}$ with a scaling factor $s$ can be expressed (prior to conversion) as follows:

$$\hat{w} = s \sum_{i=0}^{q-1} 2^i \, \hat{\mathbf{b}}_i + \hat{z}, \, \hat{\mathbf{b}}_i \in [0, 1]. \tag{5}$$

---

[1] https://github.com/NVIDIA/FasterTransformer

Then, $\hat{w}$ can be rewritten as:

$$\hat{w} = \frac{1}{2}s\sum_{i=0}^{q-1}2^i \cdot 2\hat{\mathbf{b}}_i + \hat{z} = \sum_{i=0}^{q-1}\frac{1}{2}s \cdot 2^i(2\hat{\mathbf{b}}_i - 1) + \sum_{i=0}^{q-1}\frac{1}{2}s \cdot 2^i + \hat{z}, \tag{6}$$

Given that a binary weight $b$ in the BCQ format can be defined as $b = 2\hat{b} - 1$, we obtain

$$\hat{w} = \sum_{i=0}^{q-1}(2^{i-1}s \cdot \mathbf{b}_i) + (\sum_{i=0}^{q-1}\frac{1}{2}s \cdot 2^i + \hat{z}), \tag{7}$$

which can be interpreted as a special case of BCQ, where $\alpha_i = 2^{i-1}s$ and $z = \sum_{i=0}^{q-1}\frac{1}{2}s \cdot 2^i + \hat{z} = \sum_{i=0}^{q-1}\alpha_i + \hat{z}$ in Equation 3.

In detail, to construct the binary weights and scaling factors from the pre-trained weights, we introduce a two-step methodology. Initially, scaling factors and integer weights are generated via a uniform quantization method (Frantar et al., 2022; Lin et al., 2023; Lee et al., 2023) as described in Equation 5. Subsequently, the integer weights are transformed into binary weights using bitwise operations, as described in Equation 7. More precisely, the scaling factors $\alpha_i$ and binary weights $\mathbf{b}_i$ in BCQ format for the $i$-th bit position are computed as $2^{i-1}s$ and $2\hat{\mathbf{b}}_i - 1$ respectively. Here, $s$ and $\hat{\mathbf{b}}_i$ denote the scaling factor and the binary value at the $i$-th bit position of the integer weight in uniform quantization, respectively.

## D  IMPACT ON COMPRESSION RATIO

Let $q$ be the number of quantization bits. For a given $q$, a smaller group size $g$ can lower quantization error at the expense of an increased memory footprint for scaling factors. Then, the target quantization error serves as a constraint for determining a compromise between $q$ and $g$, thus resulting in a range of achievable compression ratios. In other words, due to the introduction of $g$, we can control the amount of scaling factors and binary vectors as a trade-off process. Note that the memory footprint of conventional row-wise quantization techniques is dominated by the size of binary vectors because the size of scaling factors can usually be ignored if the column width of a matrix is large enough. Compared to the conventional scheme, our proposed group-wise BCQ provides a new wide search space for quantization formats to meet a target compression ratio.

Figure 6 shows an example with two $(g, q)$ configurations to quantize an $(4 \times 8)$ matrix. Indeed, even if the number of quantization bits is smaller, the memory footprint can be large when the group size $g$ is small (i.e., more scaling factors are employed).

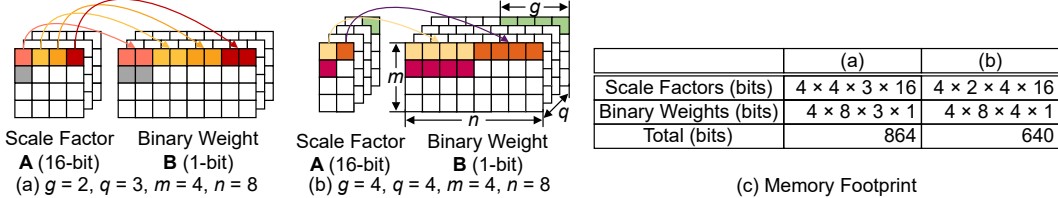

| | (a) | (b) |
|---|---|---|
| Scale Factors (bits) | 4 × 4 × 3 × 16 | 4 × 2 × 4 × 16 |
| Binary Weights (bits) | 4 × 8 × 3 × 1 | 4 × 8 × 4 × 1 |
| Total (bits) | 864 | 640 |

Scale Factor **A** (16-bit)  Binary Weight **B** (1-bit)
(a) $g$ = 2, $q$ = 3, $m$ = 4, $n$ = 8

Scale Factor **A** (16-bit)  Binary Weight **B** (1-bit)
(b) $g$ = 4, $q$ = 4, $m$ = 4, $n$ = 8

(c) Memory Footprint

Figure 6: Group-wise BCQ example with two different $(g, q)$ configurations to quantize an $(4 \times 8)$ matrix. Assuming a scaling factor of 16 bits, smaller $q$ can yield a larger memory footprint if $g$ is also small.

## E  EXPLORATION OF COMPRESSION RATIO

To study the capability of group-wise BCQ to enlarge search space for compression, we conduct experiments on three pre-trained OPT models Zhang et al. (2022), which are publicly available[2]. Specifically, we apply post-training quantization (with an iterative solver introduced in Xu et al.

---

[2]https://huggingface.co/facebook/opt-30b

(2018)) to pre-trained OPT models while $g$ and $q$ vary. Then, each quantized model is evaluated on the LAMBADA Paperno et al. (2016) dataset to find the relationship between compression ratio and accuracy. Figure 7 shows accuracy and compression ratio[3] when we try various $q$ values and $g$ values. From Figure 7, we observe that compared to the conventional row-wise quantization, group-wise BCQ offers new optimal configurations. Thus, to achieve the best compression ratio (or minimum accuracy degradation), it is necessary to explore different $q$ and $g$ values simultaneously for a given target. Note that for OPT-13B and OPT-30B, as we discussed the limits of row-wise quantization for large-scale LMs, a small $g$ value is critical to achieving low accuracy degradation (while latency is not heavily affected by a small $g$). All in all, the effects of $q$ and $g$ on accuracy differ with each model such that $q$ and $g$ are hyper-parameters to be optimized.

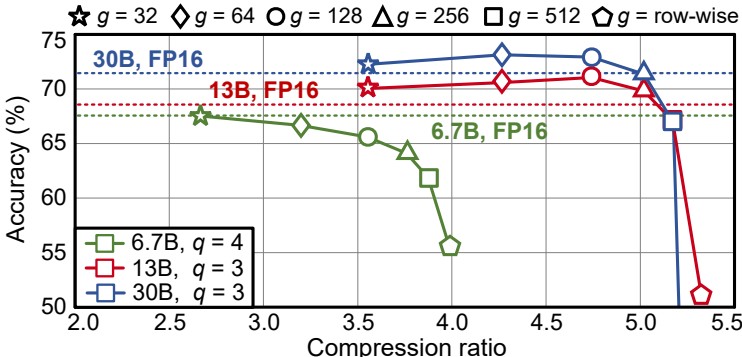

Figure 7: Accuracy and compression ratio with the various combinations of quantization bits ($q$) and group size ($g$). Three pre-trained OPT models are quantized (by post-training BCQ method) and then evaluated on the LAMBADA dataset.

## F    ADDITIONAL EXPERIMENTAL RESULTS

This section contains additional experimental results.

Table 6: Comparison of perplexity and end-to-end latency per token for OPT family models.

| Model | Kernel-$q$-$g$ | Quant. method | Perplexity | | | Latency (ms) | | |
|---|---|---|---|---|---|---|---|---|
| | | | Wiki2 | PTB | LAMBADA | 1-GPU | 2-GPU | 4-GPU |
| OPT 30B | cuBLAS-16-N/A | FP16 | 9.56 | 11.84 | 15.84 | 40.5 | 23.5 | 14.7 |
| | LUT-GEMM-4-32 | RTN | 9.71 | 12.02 | 16.16 | 18.5 | 14.3 | 11.9 |
| | LUT-GEMM-4-64 | RTN | 9.88 | 12.05 | 16.17 | 17.8 | 13.9 | 11.8 |
| | LUT-GEMM-3-32 | RTN | 10.78 | 13.89 | 19.21 | 16.7 | 13.3 | 11.6 |
| | LUT-GEMM-3-64 | RTN | 14.61 | 24.63 | 31.13 | 15.7 | 12.6 | 11.2 |
| OPT 66B | cuBLAS-16-N/A | FP16 | 9.34 | 11.36 | 15.47 | **OOM** | 48.4 | 28.3 |
| | LUT-GEMM-4-32 | RTN | 9.54 | 11.55 | 15.83 | 33.5 | 23.5 | 18.5 |
| | LUT-GEMM-4-64 | RTN | 9.54 | 11.63 | 15.91 | 31.9 | 23.2 | 17.8 |
| | LUT-GEMM-3-32 | RTN | 18.82 | 35.76 | 42.16 | 30.5 | 21.5 | 16.9 |
| | LUT-GEMM-3-64 | RTN | 51.15 | 130.54 | 130.19 | 27.5 | 20.9 | 16.0 |

---

[3]Calculated as FP16 model size divided by each quantized model size.

Table 7: Comparison of perplexity and end-to-end latency per token for LLaMA family models.

| Model | Kernel-$q$-$g$ | Quant. method | Perplexity Wiki2 | Latency (ms) 1-GPU | 2-GPU | 4-GPU |
|---|---|---|---|---|---|---|
| LLaMA 7B | cuBLAS-16-N/A | FP16 | 5.68 | 10.0 | 7.6 | 5.6 |
| | LUT-GEMM-4-128 | AWQ | 5.96 | 6.1 | 5.4 | 5.7 |
| | LUT-GEMM-3-128 | AWQ | 7.01 | 5.5 | 5.1 | 5.3 |
| LLaMA 13B | cuBLAS-16-N/A | FP16 | 5.09 | 18.2 | 12.0 | 8.6 |
| | LUT-GEMM-4-128 | AWQ | 5.25 | 10.4 | 8.0 | 6.9 |
| | LUT-GEMM-3-128 | AWQ | 5.88 | 9.3 | 7.1 | 6.4 |
| LLaMA 30B | cuBLAS-16-N/A | FP16 | 4.10 | 43.6 | 25.1 | 16.9 |
| | LUT-GEMM-4-128 | AWQ | 4.23 | 20.7 | 15.5 | 12.2 |
| | LUT-GEMM-3-128 | AWQ | 4.88 | 18.1 | 13.7 | 11.2 |
| LLaMA 65B | cuBLAS-16-N/A | FP16 | 3.53 | **OOM** | 46.9 | 29.1 |
| | LUT-GEMM-4-128 | AWQ | 3.67 | 35.7 | 25.4 | 19.9 |
| | LUT-GEMM-3-128 | AWQ | 4.24 | 31.3 | 23.0 | 18.3 |

[*] PPL numbers are extracted from the AWQ reference (Lin et al., 2023).

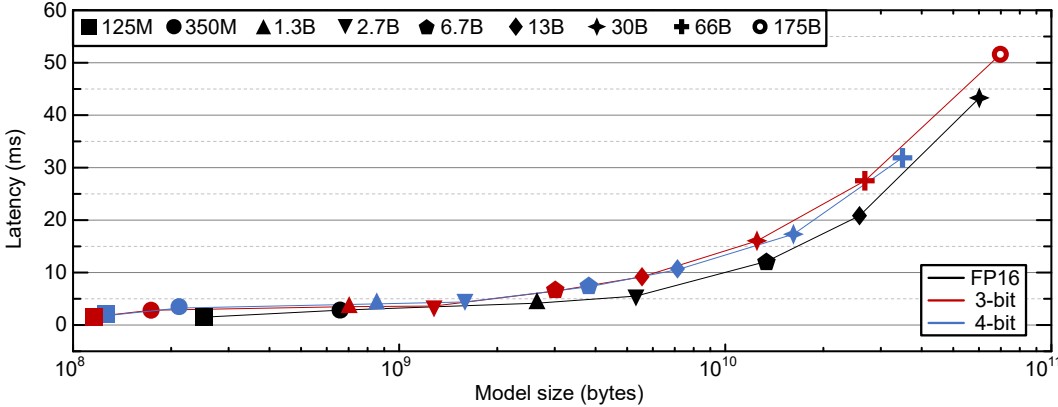

Figure 8: End-to-end latency comparison between FP16 and quantized OPT models on a single A100 80GB GPU with $g = 128$. Note that the latency data for the OPT-66B and OPT-175B models in FP16, as well as the OPT-175B model in 4-bit quantization, are omitted due to Out Of Memory (OOM) constraints.

Table 8: Accuracy achieved with 2-bit AlphaTuning on the MNLI dataset.

| Model | Method | $g$ | Acc. (%) | mm Acc. (%) |
|---|---|---|---|---|
| OPT-1.3B | Full fine-tune | - | 87.5 | 87.4 |
| | AlphaTuning | 2048 | 85.0 | 84.9 |
| | | 1024 | 85.1 | 85.1 |
| | | 256 | 85.3 | 85.6 |
| | | 128 | 85.7 | 86.2 |
| | | 64 | 86.3 | 86.6 |
| | | 32 | 86.5 | 87.1 |
| OPT-2.7B | Full fine-tune | - | 88.5 | 88.4 |
| | AlphaTuning | 1280 | 85.5 | 86.0 |
| | | 256 | 86.8 | 87.7 |
| | | 128 | 87.3 | 87.8 |
| | | 64 | 87.4 | 87.9 |
| | | 32 | 87.6 | 88.4 |

Table 9: Accuracy of quantized OPT-175B using OPTQ (without group quantization) and the corresponding end-to-end latency per token by LUT-GEMM.

| Kernel | $q$ | Perplexity | | | Accuracy (%) | Model size (GB) | Latency (ms) |
|--------|-----|------|-----|----|------|------|------|
| | | Wiki2 | PTB | C4 | PIQA | (Comp. ratio) | |
| cuBLAS | 16 | 8.34 | 12.01 | 10.13 | 81.07 | 347.9 ($\times$1.00) | 42.4 (8-GPU) |
| LUT-GEMM | 3 | 8.68 | 12.86 | 10.67 | 80.03 | 65.3 ($\times$5.33) | 51.6 (1-GPU) |

[*] PPL and accuracy numbers are extracted from the OPTQ reference (Frantar et al., 2022).

