# OpenReview forum: "LUT-GEMM: Quantized Matrix Multiplication based on LUTs for Efficient Inference in Large-Scale Generative Language Models"
_ICLR.cc/2024/Conference — ICLR 2024 poster_

### Official Review · Reviewer_kS8V · 2023-10-26

**Soundness:** 4 excellent
**Presentation:** 3 good
**Contribution:** 3 good
**Rating:** 8
**Confidence:** 5

**Summary:**

This paper proposes an interesting post-training quantization method, without needing to dequantize the integer back to FP16. In particular, they leverage binary coding quantization (BCQ) to represent the LLM weights in both uniform and non-uniform fashion, resulting in a set of scaling factor matrices A and binary weight matrices B of +1 or -1. Therefore, the multiplication between weights W and activations X will become A(BX), where BX is composed of repetitive additions with partial sums being stored on the buffer as LUT to accelerate the computation.

The author also provides a customized CUDA kernel to deploy such LUT-based GEMM on GPUs, achieving a 2.1× improvement on a single GPU when compared to OPTQ using OPT-175B models with 3-bit weight-only quantization.

**Strengths:**

1. Clear ideas and descriptions of representing weights with scaling factor matrices and binary weight matrices. The advantage of using LUT and grouped scaling factors is extensively evaluated. The way of leveraging fine-grained scaling factors enabled a non-uniform quantization format.

2. Solid implementation. The analysis and GPU implementation stand this idea out. I understand the importance of having it work on GPUs but I still wonder how efficient is the implemented kernel as compared to the customized hardware accelerator.

3. Plausible ablation studies. There are a lot of ablation studies to show the effectiveness of every proposed component.

**Weaknesses:**

One big issue in writing that prevents one from reimplementing the proposed method is how to construct the binary matrices and scaling factor matrices from the pre-trained weights. The attached code assumes randomized matrices so that part cannot provide any useful information regarding this.

For more technical questions:

1. Regarding the LUT part, what is the repetition that we can exploit form? The quantitative measurements or ablation studies on this will provide more insights into how many benefits we can get for such LUT-based GEMM instead of vanilla additions. Any theoretical analysis on the upper bound of savings brought from LUT?

2. Regarding the scaling factor part, how to determine the group size? In my understanding, if the group size is small, there are no benefits as you have as many matrices as used bits to represent weights. At what group size, do the benefits compensate the cost of the scaling factor matrices? Is there any formula to calculate this?

3. Regarding the experiments, why only consider the OPT model? How about the speedups on other candidates like LLaMa at different scales? In that way, we can also understand the scalability of the used kernel better.

**Questions:**

See weaknesses.

In addition, the authors did not clearly describe how you derive the scaling factors matrices and binary matrices from the pre-trained weights. I think that part is necessary and it will be better if there is any theoretical analysis of the quantization error or variance.

I am open to raising the score depending on the good answers.

--
Update: Thank you for the comprehensive response. Many of my concerns have been resolved, and I've increased the rating score.

---

> ### Author Response · Authors · 2023-11-18
>
> We highly appreciate your kind efforts in evaluating our work. The manuscript has been carefully
> revised to address your comments as below.
>
> ## Weakness 0 & Question
>
> > One big issue in writing that prevents one from reimplementing the proposed method is how to construct the binary matrices and scaling factor matrices from the pre-trained weights. The attached code assumes randomized matrices so that part cannot provide any useful information regarding this.
>
> > In addition, the authors did not clearly describe how you derive the scaling factors matrices and binary matrices from the pre-trained weights. I think that part is necessary and it will be better if there is any theoretical analysis of the quantization error or variance.
>
> According to this comment, we have included an example implementation in the supplementary material demonstrating the construction of binary matrices and scaling factor matrices from pre-trained weights. Please refer to the examples directory provided in the supplementary material.
>
> In the case of uniform quantization, our process involves initially obtaining scaling factor matrices and integer weight matrices through a uniform quantization method (we utilized the simple RTN method in our example). Subsequently, we proceed to convert the integer weight matrix into binary weight matrices using bitwise operations, as explained in detail in Appendix C of the revised manuscript. Additionally, to form a key for LUT-GEMM, we pack eight consecutive binary weights in a row into a uint8 data type. All these steps are implemented in the example directory.
>
> In the revised manuscript, we’ve clearly highlighted how to construct the binary matrices and scaling factor matrices from the pre-trained weights.
> Thank you for your feedback to improve the quality of our manuscript significantly.
>
>
> ## Weakness 1
>
> > Regarding the LUT part, what is the repetition that we can exploit form? The quantitative measurements or ablation studies on this will provide more insights into how many benefits we can get for such LUT-based GEMM instead of vanilla additions. Any theoretical analysis on the upper bound of savings brought from LUT?
>
> As you mentioned, there is an upper bound to the benefits achievable when using LUT-GEMM, and in certain scenarios, it might be less efficient than a typical GEMM approach. When employing LUT-GEMM, the advantage lies in precomputing all additions and subtractions of inputs $x_0$ to $x_7$ through a 256-size table, reducing the original 8 operations to just 1 operation. This translates to a reduction from 8 operations to 1, resulting in a 1/8 computation load. However, this applies to the 1-bit scenario; in the case of 4 bits, there is a 4/8, or 1/2, computational advantage if the memory bandwidth is sufficiently large. Nonetheless, due to limitations in memory bandwidth, the reduction in memory (from 16 bits to 4 bits) theoretically yields a fourfold maximum speed improvement cost-effectively.
>
> However, there are limitations even with LUT-GEMM. In cases where the *hidden_dim* is exceptionally small, the cost of creating a lookup table might outweigh the computational cost, leading to a relative decrease in speed enhancement. This scenario occurs when *hidden_dim* is smaller than 512, a situation rarely seen in LLM cases.

---

> > ### Author Response · Authors · 2023-11-18
> >
> > ## Weakness 2
> >
> > > Regarding the scaling factor part, how to determine the group size? In my understanding, if the group size is small, there are no benefits as you have as many matrices as used bits to represent weights. At what group size, do the benefits compensate the cost of the scaling factor matrices? Is there any formula to calculate this?
> >
> > Thank you for your insightful comment. We recognize the impact of group size on memory footprint, particularly how scale factors contribute to increased memory usage when the group size decreases. We visualize the formula for calculating the overall memory footprint as a function of both quantization bit and group size in Figure 6.
> >
> > The determination of the group size typically involves heuristic approaches to identify an optimal point that meets system requirements effectively. This choice significantly influences various aspects such as model size, end-to-end latency, and accuracy. It's worth noting that a commonly adopted group size value is g = 128, as reported in [1-3], owing to its favorable balance across these critical system parameters.
> >
> > We appreciate your attention to this aspect and hope that our visual representation in Figure 6 provides clarity on the relationship between quantization bit, group size, and memory footprint in our proposed methodology.
> >
> > Moreover, in Table 5 of our manuscript, we aimed to present the latency and accuracy metrics for the quantized OPT-175B model, specifically compressed for deployment on a single GPU. An interesting observation emerges when comparing the configurations (q, g)=(2, 32) and (q, g)=(3, -). Despite the smaller model size associated with (q, g)=(2, 32), there is an evident increase in end-to-end latency compared to (q, g)=(3, -). This anomaly is primarily attributed to the overhead latency incurred in smaller group sizes, as illustrated in Figure 4(a).
> >
> > This discrepancy highlights the influence of group size on latency, where a smaller group size can introduce additional overhead despite reducing the model size. We appreciate your understanding and hope that this clarification resolves the complexities associated with latency variations across various configurations of quantization bits and group sizes.
> >
> > [1] Lin, Ji, et al. "AWQ: Activation-aware Weight Quantization for LLM Compression and Acceleration." arXiv preprint arXiv:2306.00978 (2023). \
> > [2] Frantar, Elias, et al. "OPTQ: Accurate quantization for generative pre-trained transformers." The Eleventh International Conference on Learning Representations. 2022. \
> > [3] Kim, Sehoon, et al. "SqueezeLLM: Dense-and-Sparse Quantization." arXiv preprint arXiv:2306.07629 (2023).
> >
> >
> > ## Weakness 3
> >
> > > Regarding the experiments, why only consider the OPT model? How about the speedups on other candidates like LLaMa at different scales? In that way, we can also understand the scalability of the used kernel better.
> >
> > To take this comment into account, we have expanded our evaluations to include latency assessments across the LLaMA model family (Table 7 in the revised manuscript).
> >
> > |           | Kernel-q-g     | Quant. Method | \*PPL | 1-GPU   | 2-GPU   | 4-GPU   |
> > | --------- | -------------- | ------------- | ----- | ------- | ------- | ------- |
> > | LLaMA-7B  | cuBLAS-16-N/A  | FP16          | 5.68  | 9.99956 | 7.64547 | 5.58484 |
> > | LLaMA-7B  | LUT-GEMM-4-128 | RTN           | 5.96  | 6.10075 | 5.41022 | 5.70228 |
> > | LLaMA-7B  | LUT-GEMM-3-128 | RTN           | 7.01  | 5.48141 | 5.10903 | 5.27691 |
> > | LLaMA-13B | cuBLAS-16-N/A  | FP16          | 5.09  | 18.2208 | 11.969  | 8.57203 |
> > | LLaMA-13B | LUT-GEMM-4-128 | RTN           | 5.25  | 10.3895 | 8.02672 | 6.90163 |
> > | LLaMA-13B | LUT-GEMM-3-128 | RTN           | 5.88  | 9.28256 | 7.14944 | 6.35469 |
> > | LLaMA-30B | cuBLAS-16-N/A  | FP16          | 4.1   | 43.5962 | 25.0504 | 16.9111 |
> > | LLaMA-30B | LUT-GEMM-4-128 | RTN           | 4.23  | 20.6537 | 15.5217 | 12.1708 |
> > | LLaMA-30B | LUT-GEMM-3-128 | RTN           | 4.88  | 18.1275 | 13.6595 | 11.2222 |
> > | LLaMA-65B | cuBLAS-16-N/A  | FP16          | 3.53  | OOM     | 46.8719 | 29.1132 |
> > | LLaMA-65B | LUT-GEMM-4-128 | RTN           | 3.67  | 35.6644 | 25.4327 | 19.852  |
> > | LLaMA-65B | LUT-GEMM-3-128 | RTN           | 4.24  | 31.2953 | 22.9804 | 18.342  |
> >
> > \* PPL from AWQ reference [1] \
> > [1] Lin, Ji, et al. "AWQ: Activation-aware Weight Quantization for LLM Compression and Acceleration." arXiv preprint arXiv:2306.00978 (2023).

---

### Official Review · Reviewer_banf · 2023-10-30

**Soundness:** 3 good
**Presentation:** 3 good
**Contribution:** 2 fair
**Rating:** 6
**Confidence:** 4

**Summary:**

This paper proposes an quantization optimization method for Transformer-like model on GEMM calculation, which uses integer weight quantization and floating-point activation values, and uses a lookup table computation scheme to achieve inference acceleration. The paper provides specific GPU implementation methods and performance analysis, explains the implementation methods of uniform quantization and non-uniform quantization in this scheme. In the experimental section, the author compared with the basic cuBLAS implementation, and the results showed that the proposed method has much better performance and does not cause significant loss of inference accuracy.

**Strengths:**

1. The work of this paper is of great significance, especially for optimizing the inference performance of LLM, which is currently highly concerned. Compared to the baseline cuBLAS implementation, the performance of this method has significantly improved.
2. The design, experiment, and analysis of this paper are relatively solid. For the algorithm design of GPU, this paper provides detailed explanations and time complexity analysis. This paper carefully considers the performance of different compression ratios under different bit and grouping parameters g.
3. CUDA kernel code and performance test code is provided.

**Weaknesses:**

1. The novelty of this paper is ordinary. The main parts, including BCQ quantification and group-wise quantization optimization, both come from existing research. This paper proposes to convert uniform quantization into BCQ format, which is not a complex transformation.
2. The author mentioned that the optimization here currently only focuses on the inference of a single batch, which may limit its use.
3. The LUT-Based method requires a significant amount of memory capacity in exchange for efficiency. In Table 2, the 4-bit quantified LUT-GEMM storage footprint exceeds the 16 bit model of the cuBLAS baseline. In fact, storage resources are also the main focus of quantization in large language models, not just performance. This paper seems to focus mainly on computational efficiency, but lacks a comparison between memory resource usage.
4. The experimental data in this paper is not enough, and the main baseline for comparison is 16 bit cuBLAS, lacking a comprehensive comparison of other quantization methods. The experiment only tested the quantization accuracy of PPL indicators using the OPT model on the LAMBADA dataset, which cannot fully demonstrate the effectiveness and universality of the method.

**Questions:**

1. The speed of inference is important, but it is only one part. I hope the author can compare the advantages and disadvantages of other quantization acceleration methods, including compression ratio of GPU memory requirements, the need for post training calibration, etc.
2. The author claims that the BCQ quantization used is suitable for both uniform and non uniform quantization, but the experiment only considered uniform quantization. Theoretically, non-uniform quantization has better representation ability to achieve higher accuracy. Although testing the performance of non-uniform quantization may be difficult on current hardware, it is still possible to compare the accuracy differences between the two methods.

---

> ### Author Response · Authors · 2023-11-17
>
> We deeply appreciate the invaluable feedback provided by the reviewer.
>
> ## Weakness 1
>
> > The novelty of this paper is ordinary. The main parts, including BCQ quantification and group-wise quantization optimization, both come from existing research. This paper proposes to convert uniform quantization into BCQ format, which is not a complex transformation.
>
> Thanks for your careful comment. In addressing the raised concern regarding the originality of our work, it is crucial to highlight the distinctive contributions of LUT-GEMM, which closely align with recent research trends.
> - LUT-GEMM showcases compatibility with cutting-edge model compression techniques. Current investigations into weight-only quantization for Large Language Model (LLM) compression predominantly emphasize uniform quantization [1, 2]. In Appendix C, we clearly demonstrate that LUT-GEMM is not only applicable  to non-uniformly quantized models but also to uniformly quantized models, which is neglected in [3] . This showcases the versatility of LUT-GEMM in harnessing the effects of cutting-edge quantization technologies.
> - LUT-GEMM introduces group-wise quantization, offering a novel search space for LUT-based kernels. This facilitates the fine-tuning of the trade-off between model size, latency, and accuracy. The flexibility provided by group-wise quantization enables the customization of systems to meet the specific requirements of diverse services.
> - In contrast to the focus of prior work [3] on CPU optimization, which demonstrated suboptimal performance on GPUs compared to cuBLAS, LUT-GEMM provides tailored optimization and a comprehensive analysis for the most widely used Large Language Model (LLM) accelerator—GPUs. Our proposed method exhibits superior performance on GPUs compared to cuBLAS, and we offer a thorough analysis regarding optimization on GPUs.
>
> [1] Lin, Ji, et al. "AWQ: Activation-aware Weight Quantization for LLM Compression and Acceleration." arXiv preprint arXiv:2306.00978 (2023). \
> [2] Frantar, Elias, et al. "OPTQ: Accurate quantization for generative pre-trained transformers." The Eleventh International Conference on Learning Representations. 2022. \
> [3] Jeon, Yongkweon, et al. "Biqgemm: matrix multiplication with lookup table for binary-coding-based quantized dnns." SC20: International Conference for High Performance Computing, Networking, Storage and Analysis. IEEE, 2020.
>
> ## Weakness 2
>
> > The author mentioned that the optimization here currently only focuses on the inference of a single batch, which may limit its use.
>
> In Section 6, we acknowledge that LUT-GEMM exhibits diminishing performance gains with increasing batch size. This phenomenon arises due to the effective utilization of a potent built-in GEMM accelerator Tensor core, which becomes operational when the batch size exceeds 4. Consequently, the design choice of LUT-GEMM, employing CUDA core for single batch operations, is intentionally tailored for scenarios assumed to process single requests at a time in personal computers, prioritizing privacy considerations.
>
> ## Weakness 3
>
> > The LUT-Based method requires a significant amount of memory capacity in exchange for efficiency. In Table 2, the 4-bit quantified LUT-GEMM storage footprint exceeds the 16 bit model of the cuBLAS baseline. In fact, storage resources are also the main focus of quantization in large language models, not just performance. This paper seems to focus mainly on computational efficiency, but lacks a comparison between memory resource usage.
>
> We appreciate your feedback, and we acknowledge that there were some confusing descriptions in the original manuscript regarding memory utilization in Table 2. To clarify, the presented memory utilization in Table 2 represents the percentage of time over the past sample period during which global (device) memory was being read or written [4]. In essence, the memory utilization metric is closely linked to memory movement rather than the amount of memory used in a GPU.
>
> In reality, the memory footprint of LUT-GEMM is significantly smaller than that of cuBLAS. This substantial reduction in memory usage allows us to successfully execute a 175B model on a single GPU, as demonstrated in Table 4. In contrast, attempting to run the 16-bit model of the cuBLAS baseline results in an out-of-memory (OOM) occurrence. We have taken your feedback into consideration and, in the revised manuscript, have made efforts to clearly present the definition of memory utilization in Table 2 to avoid any confusion.
>
> [4] https://nvidia.custhelp.com/app/answers/detail/a_id/3751/~/useful-nvidia-smi-queries

---

> > ### Author Response · Authors · 2023-11-17
> >
> > ## Weakness 4
> >
> > > The experimental data in this paper is not enough, and the main baseline for comparison is 16 bit cuBLAS, lacking a comprehensive comparison of other quantization methods. The experiment only tested the quantization accuracy of PPL indicators using the OPT model on the LAMBADA dataset, which cannot fully demonstrate the effectiveness and universality of the method.
> >
> > We are sorry that there were some confusing descriptions in the original manuscript. LUT-GEMM is designed to be agnostic to the specific quantization method employed. It can seamlessly support any quantization method that can be represented in the form outlined in Equation 3. To address this concern, we have conducted additional evaluations of latency on the LLaMA model family, employing the baseline quantization method, RTN (Table 7 in the revised manuscript). This extended evaluation provides a more comprehensive understanding of the versatility and compatibility of LUT-GEMM across various quantization approaches.
> >
> > ## Question 1
> >
> > > The speed of inference is important, but it is only one part. I hope the author can compare the advantages and disadvantages of other quantization acceleration methods, including compression ratio of GPU memory requirements, the need for post training calibration, etc.
> >
> > Thank you for your feedback. It's important to highlight that the memory footprint of LUT-GEMM exhibits a significant reduction compared to cuBLAS, enabling the successful execution of a 175B model on a single GPU. Similarly, other quantization acceleration such as  AWQ [1] and OPTQ [2] methods successfully execute larger models on a single GPU.
> >
> > One of the notable advantages of LUT-GEMM lies in its quantization bit agnosticism. Specifically, the OPTQ kernel, optimized for 3-bit quantized weights, utilizes three 32-bit data types to efficiently store the quantized weights. Similarly, the AWQ kernel, optimized for 4-bit quantized weights, employs a single 32-bit data type for efficient weight storage. Unlike previous dequantization-based kernels that require distinct kernels for different quantization bit configurations, LUT-GEMM operates on each binary weight matrix for every bit position. Consequently, LUT-GEMM possesses the capability to support multiple quantization configurations within a single kernel, enhancing its versatility and reducing the need for multiple specialized kernels.
> >
> > [1] Lin, Ji, et al. "AWQ: Activation-aware Weight Quantization for LLM Compression and Acceleration." arXiv preprint arXiv:2306.00978 (2023). \
> > [2] Frantar, Elias, et al. "OPTQ: Accurate quantization for generative pre-trained transformers." The Eleventh International Conference on Learning Representations. 2022.
> >
> > ## Question 2
> >
> > > The author claims that the BCQ quantization used is suitable for both uniform and non uniform quantization, but the experiment only considered uniform quantization. Theoretically, non-uniform quantization has better representation ability to achieve higher accuracy. Although testing the performance of non-uniform quantization may be difficult on current hardware, it is still possible to compare the accuracy differences between the two methods.
> >
> > As you pointed out with this comment, uniform quantization methods often exhibit a drawback by allocating excessive precision to the edges of the distribution, making them generally show worse accuracy compared to non-uniform quantization methods. Despite this disadvantage, the active exploration of uniform quantization methods persists, primarily driven by their ease of deployment without requiring specialized kernels. Our contribution lies in the development of an open-source library for LUT-GEMM, which supports both uniform and non-uniform quantization methods. By providing this flexibility, we aim to encourage a shift towards highlighting non-uniform quantization methods which have not been actively explored in the research community due to the lack of practical supporting kernels. We anticipate that this emphasis will contribute to the evolution of more accurate quantization techniques in the future, leveraging the versatility of our library to facilitate advancements in the field.

---

> > > ### Comment · Reviewer_banf · 2023-11-20
> > >
> > > Thank you for your valuable response and clarification. I am glad to see that LUT-GEMM has effectively compressed storage space and achieved competitive results on the LLaMA model. Additionally, 4-bit AWQ or OPTQ storage compression comparisons can be added to Table 5 or other tables, rather than just having a cuBLAS 16-bit baseline.
> > >
> > > Although I believe that paper has only made minor or moderate technical innovations to existing BCQ, it still has practical significance in facing LLM scenarios.
> > >
> > > About the inference limitations of one or small batchsize, the single user scenario is not sufficient to indicate that there is only one batch, and there are some acceleration method such as Speculative Decoding [1], which may be a problem to consider in the future.
> > >
> > > Anyway, your explanation covers some of my main concerns. The method exhibits competitive results compared to the SOTA method in terms of generality and inference speed. I will change the rating from 5 to 6.
> > >
> > > [1] Leviathan, Y., Kalman, M.,&Matias, Y. (2022) Fast Inference from Transformers via Speculative Decoding International Conference on Machine Learning

---

> > > > ### Author Response · Authors · 2023-11-21
> > > >
> > > > We appreciate your feedback and are pleased that the explanations provided have helped address some of your primary concerns. Following your suggestion, we have included a comparison of 4-bit OPTQ storage compression in Table 5 of the revised manuscript. Your insight regarding the inference limitations associated with small batch sizes is well noted. The point about a single-user scenario not necessarily representing only one batch is valid and insightful. Additionally, your mention of acceleration methods like Speculative Decoding, which could be pivotal for future considerations, is both relevant and appreciated.
> > > >
> > > > Your feedback and insights are valuable in refining and clarifying the manuscript. Thank you for your detailed assessment and constructive comment.

---

### Official Review · Reviewer_UZqJ · 2023-10-31

**Soundness:** 3 good
**Presentation:** 3 good
**Contribution:** 3 good
**Rating:** 6
**Confidence:** 4

**Summary:**

This paper proposes to replace FP16 GEMM calls in weight-only-quantization neural network inference with look-up-table-based GEMMs. The proposed technique is designed to help scenarios where the bit width of weight-only-quantization is 4 or less and the weight matrix size is large enough, as is the case for large language models. The core idea is that, when the bit width of weights is low, one can pre-compute multiplication results with the activation tensor for all combinations and store them, and hence most of the multiplications are replace by table look up; the amount of sum operations remain about the same; the amount of memory reads remain about the same. Experimental results include kernel-level latency comparison in Table 2 and end-to-end latency comparison on OPT in Tables 3-5. Source code is included in the supplementary.

**Strengths:**

The core idea of the paper makes sense and in fact is more general and beyond the BCQ format used this paper. The key question on rating the originality is the relation to prior works. For example, how does this work relates to BiQGEMM (Jeon et al., 2020)? Section 3.1 does acknowledge this and a few other prior works but it's unclear what are the distinctions. If a satisfactory answer could be provided, then originality could be a big strength for this paper.

The experimental setup is good, particularly that perplexity test is included.

Including source code is a big plus, although I have not verified it.

**Weaknesses:**

As said above, the originality question is the key. Please provide detailed narrative on the differentiation. Including prior kernels in evaluations would be even stronger.

The last row of Table 2 suggests 4X speed up with 4-bit quantization at kernel level. However this does not seems to translate to the 4-bit end2end latency in Tables 3 and 4, not anywhere close to 4X. Can you explain why?

A critical property of the proposal is that the benefit gets larger as the matrix dimension increases. Right now the results only show end2end latency for OPT 66B and 175B. Could you add a plot of latency as a function of OPT model sizes, all the way from the smallest to 175B?

**Questions:**

Please see the weaknesses section.

---

> ### Author Response · Authors · 2023-11-17
>
> We express our sincere gratitude for the invaluable feedback offered by the reviewer.
>
> ## Weakness 1
>
> > As said above, the originality question is the key. Please provide detailed narrative on the differentiation. Including prior kernels in evaluations would be even stronger.
>
> Thanks for your careful comment. In addressing the raised concern regarding the originality of our work, it is crucial to highlight the distinctive contributions of LUT-GEMM, which closely align with recent research trends.
> - LUT-GEMM showcases compatibility with cutting-edge model compression techniques. Current investigations into weight-only quantization for Large Language Model (LLM) compression predominantly emphasize uniform quantization [1, 2]. In Appendix C, we clearly demonstrate that LUT-GEMM is not only applicable to non-uniformly quantized models but also to uniformly quantized models, which is neglected in [3]. This showcases the versatility of LUT-GEMM in harnessing the effects of cutting-edge quantization technologies.
> - LUT-GEMM introduces group-wise quantization, offering a novel search space for LUT-based kernels. This facilitates the fine-tuning of the trade-off between model size, latency, and accuracy. The flexibility provided by group-wise quantization enables the customization of systems to meet the specific requirements of diverse services.
> - In contrast to the focus of prior work [3] on CPU optimization, which demonstrated suboptimal performance on GPUs compared to cuBLAS, LUT-GEMM provides tailored optimization and a comprehensive analysis for the most widely used Large Language Model (LLM) accelerator—GPUs. Our proposed method exhibits superior performance on GPUs compared to cuBLAS, and we offer a thorough analysis regarding optimization on GPUs.
>
> [1] Lin, Ji, et al. "AWQ: Activation-aware Weight Quantization for LLM Compression and Acceleration." arXiv preprint arXiv:2306.00978 (2023). \
> [2] Frantar, Elias, et al. "OPTQ: Accurate quantization for generative pre-trained transformers." The Eleventh International Conference on Learning Representations. 2022. \
> [3] Jeon, Yongkweon, et al. "Biqgemm: matrix multiplication with lookup table for binary-coding-based quantized dnns." SC20: International Conference for High Performance Computing, Networking, Storage and Analysis. IEEE, 2020.
>
> ## Weakness 2
>
> > The last row of Table 2 suggests 4X speed up with 4-bit quantization at kernel level. However this does not seems to translate to the 4-bit end2end latency in Tables 3 and 4, not anywhere close to 4X. Can you explain why?
>
> We sincerely appreciate your valuable feedback on our manuscript. Upon careful reevaluation, we've identified an error in the reported experimental results regarding the speedup achieved by our proposed method. In our manuscript, we originally stated a speedup of 4.32x in matrix multiplication, but upon revisiting our findings, we acknowledge that the actual speedup achieved was 3.23x. Now, the result is aligned with our end-to-end latency evaluation represented in Table 4. We deeply regret this oversight and recognize the importance of accurately representing our experimental outcomes. To rectify this mistake, we revise the manuscript to reflect the correct speedup, ensuring the accuracy and integrity of our reported findings. Once again, we extend our gratitude for your meticulous review, which has enabled us to address this error and enhance the precision of our work.
>
> Furthermore, to demonstrate the effect of speed up in matrix multiplication to end-to-end performance, we have conducted latency evaluations on the OPT-30B model. Note that, for a fair comparison with FP16 cuBLAS, we utilized the OPT-30B model, which is executable on a single GPU without quantization. As illustrated in the table below, the end-to-end speedup closely correlates with the speedup achieved in matrix multiplication.
>
> | Kernel        |  MatMul latency (3d$\times$d)  | MatMul latency (4d$\times$d)   | MatMul latency (d$\times$4d)   | End-to-end latency          |
> | --------------- | --------------- | --------------- | --------------- | ------------- |
> | cuBLAS (FP16)    | 0.19544ms         | 0.26811ms         | 0.25943ms         | 43.28ms         |
> | LUT-GEMM (4bit) | 0.08903ms ($\times$2.20) | 0.10920ms ($\times$2.46) | 0.11334ms ($\times$2.29) | 17.34ms ($\times$2.49) |
> | LUT-GEMM (3bit) | 0.07728ms ($\times$2.53) | 0.09336ms ($\times$2.87) | 0.09471ms ($\times$2.74) | 16.04ms ($\times$2.70) |
>
> ## Weakness 3
>
> > A critical property of the proposal is that the benefit gets larger as the matrix dimension increases. Right now the results only show end2end latency for OPT 66B and 175B. Could you add a plot of latency as a function of OPT model sizes, all the way from the smallest to 175B?
>
> The authors highly appreciate this careful suggestion. During the revision, we have added a figure in the revised manuscript to show end-to-end latency as a function of OPT model sizes (Figure 8).

---

> ### Comment · Reviewer_UZqJ · 2023-11-22
> **thanks for the updates**
>
> I have read the rebuttal and thanks for the correction and additional data and explanation. My rating is unchanged.

---

> > ### Author Response · Authors · 2023-11-23
> >
> > Thank you for reviewing our rebuttal. We appreciate your time and consideration.

---

### Official Review · Reviewer_cNRT · 2023-11-03

**Soundness:** 4 excellent
**Presentation:** 4 excellent
**Contribution:** 3 good
**Rating:** 8
**Confidence:** 4

**Summary:**

Binary Code Quantization (BCQ) applied to large language model weights by LUT-GEMM marks an innovative step, where weight decomposition into quantized matrices and scaling factors is an offline process. This offers a flexibility to balance between compression and accuracy by varying group sizes and quantization bits. At execution time, activations are split into sub-vectors with corresponding look-up tables for sub-sum calculations, replacing the typical multiply-accumulate operations. This "multiplication" of binary matrices with activations, followed by scaling, optimizes the model's runtime efficiency. UT-GEMM's application on well-known models, including GPT-3, illustrates significant enhancements in performance, memory usage, energy consumption, and a reduction in necessary GPUs.

**Strengths:**

Extensive empirical evidence presented through various data visualizations confirms the efficiency of LUT-GEMM method.

The research goes further than most in addressing the real-world deployment challenges on standard hardware, which is a standout feature.

Impressive experiment section focusing on the largest open-source models and fitting them onto a single GPU.

Detailed latency insights provided in the paper lay a solid foundation for understanding the efficiency gains.

Well-written motivation and related work sections. Discussions show a deep understanding of GPU programming.

**Weaknesses:**

The achievement of deploying huge models on a singular GPU is undermined by the lack of reported accuracy metrics at this scale.

The paper lacks a direct accuracy and latency comparison among different models, although the OPT model family is a good choice.

The paper does seem to have some limited technical novelty since it adapts and extends prior method in minor to moderate ways, but the analysis and implementation seem to make up for it.

**Questions:**

How can the method be extended to larger batches or scenarios when multiple single requests are batched into one?

Is the ability to represent uniform quantization that important in this method? Uniform methods are typically strictly worse as they over-allocate precision to the edges of the distribution.

---

> ### Author Response · Authors · 2023-11-17
>
> We are grateful for the valuable feedback provided by the reviewer. In this response, we thoroughly address each of your comments and suggestions.
>
> ## Weakness 1
>
> >The achievement of deploying huge models on a singular GPU is undermined by the lack of reported accuracy metrics at this scale.
>
> We apologize for the unclear descriptions. In Table 5 of our manuscript, we aimed to present the latency and accuracy metrics for the quantized OPT-175B model, specifically compressed for deployment on a single GPU. It's worth noting that the PPL, where lower values indicate better performance, exhibits an increase as the quantization bit $q$ increases and the group size g decreases. In the revised manuscript, we have included a comprehensive accuracy evaluation at $q=3$ with newly added Table 9.
>
> ## Weakness 2
>
> >The paper lacks a direct accuracy and latency comparison among different models, although the OPT model family is a good choice.
>
> To take this comment into account, we have expanded our evaluations to include latency assessments across the LLaMA model family (Table 7 in the revised manuscript).
>
> |           | Kernel-q-g     | Quant. Method | \*PPL | 1-GPU   | 2-GPU   | 4-GPU   |
> | --------- | -------------- | ------------- | ----- | ------- | ------- | ------- |
> | LLaMA-7B  | cuBLAS-16-N/A  | FP16          | 5.68  | 9.99956 | 7.64547 | 5.58484 |
> | LLaMA-7B  | LUT-GEMM-4-128 | RTN           | 5.96  | 6.10075 | 5.41022 | 5.70228 |
> | LLaMA-7B  | LUT-GEMM-3-128 | RTN           | 7.01  | 5.48141 | 5.10903 | 5.27691 |
> | LLaMA-13B | cuBLAS-16-N/A  | FP16          | 5.09  | 18.2208 | 11.969  | 8.57203 |
> | LLaMA-13B | LUT-GEMM-4-128 | RTN           | 5.25  | 10.3895 | 8.02672 | 6.90163 |
> | LLaMA-13B | LUT-GEMM-3-128 | RTN           | 5.88  | 9.28256 | 7.14944 | 6.35469 |
> | LLaMA-30B | cuBLAS-16-N/A  | FP16          | 4.1   | 43.5962 | 25.0504 | 16.9111 |
> | LLaMA-30B | LUT-GEMM-4-128 | RTN           | 4.23  | 20.6537 | 15.5217 | 12.1708 |
> | LLaMA-30B | LUT-GEMM-3-128 | RTN           | 4.88  | 18.1275 | 13.6595 | 11.2222 |
> | LLaMA-65B | cuBLAS-16-N/A  | FP16          | 3.53  | OOM     | 46.8719 | 29.1132 |
> | LLaMA-65B | LUT-GEMM-4-128 | RTN           | 3.67  | 35.6644 | 25.4327 | 19.852  |
> | LLaMA-65B | LUT-GEMM-3-128 | RTN           | 4.24  | 31.2953 | 22.9804 | 18.342  |
>
> \* PPL from AWQ reference [1] \
> [1] Lin, Ji, et al. "AWQ: Activation-aware Weight Quantization for LLM Compression and Acceleration." arXiv preprint arXiv:2306.00978 (2023).
>
> ## Weakness 3
>
> >The paper does seem to have some limited technical novelty since it adapts and extends prior method in minor to moderate ways, but the analysis and implementation seem to make up for it.
>
>
> Thanks for your careful comment. In addressing the raised concern regarding the originality of our work, it is crucial to highlight the distinctive contributions of LUT-GEMM, which closely align with recent research trends.
> - LUT-GEMM showcases compatibility with cutting-edge model compression techniques. Current investigations into weight-only quantization for Large Language Model (LLM) compression predominantly emphasize uniform quantization [1, 2]. In Appendix C, we clearly demonstrate that LUT-GEMM is not only applicable to non-uniformly quantized models but also to uniformly quantized models, which is neglected in [3]. This showcases the versatility of LUT-GEMM in harnessing the effects of cutting-edge quantization technologies.
> - LUT-GEMM introduces group-wise quantization, offering a novel search space for LUT-based kernels. This facilitates the fine-tuning of the trade-off between model size, latency, and accuracy. The flexibility provided by group-wise quantization enables the customization of systems to meet the specific requirements of diverse services.
> - In contrast to the focus of prior work [3] on CPU optimization, which demonstrated suboptimal performance on GPUs compared to cuBLAS, LUT-GEMM provides tailored optimization and a comprehensive analysis for the most widely used Large Language Model (LLM) accelerator—GPUs. Our proposed method exhibits superior performance on GPUs compared to cuBLAS, and we offer a thorough analysis regarding optimization on GPUs.
>
> [1] Lin, Ji, et al. "AWQ: Activation-aware Weight Quantization for LLM Compression and Acceleration." arXiv preprint arXiv:2306.00978 (2023). \
> [2] Frantar, Elias, et al. "OPTQ: Accurate quantization for generative pre-trained transformers." The Eleventh International Conference on Learning Representations. 2022. \
> [3] Jeon, Yongkweon, et al. "Biqgemm: matrix multiplication with lookup table for binary-coding-based quantized dnns." SC20: International Conference for High Performance Computing, Networking, Storage and Analysis. IEEE, 2020.

---

> > ### Author Response · Authors · 2023-11-17
> >
> > ## Question 1
> >
> > > How can the method be extended to larger batches or scenarios when multiple single requests are batched into one?
> >
> > As mentioned in Section 6, we acknowledge that LUT-GEMM exhibits diminishing performance gains with increasing batch size. This phenomenon arises due to the effective utilization of a potent built-in GEMM accelerator Tensor core, which becomes operational when the batch size exceeds 4. Consequently, the design choice of LUT-GEMM, employing CUDA core for single batch operations, is intentionally tailored for scenarios assumed to process single requests at a time, for example, in personal computers, prioritizing privacy considerations.
> >
> > ## Question 2
> >
> > > Is the ability to represent uniform quantization that important in this method? Uniform methods are typically strictly worse as they over-allocate precision to the edges of the distribution.
> >
> > As you pointed out with this comment, uniform quantization methods often exhibit a drawback by allocating excessive precision to the edges of the distribution, making them generally show worse accuracy compared to non-uniform quantization methods. Despite this disadvantage, the active exploration of uniform quantization methods persists [1, 2], primarily driven by their ease of deployment without requiring specialized kernels. Our contribution lies in the development of an open-source library for LUT-GEMM, which supports both uniform and non-uniform quantization methods. By providing this flexibility, we aim to encourage a shift towards highlighting non-uniform quantization methods which have not been actively explored in the research community due to the lack of practical supporting kernels. We anticipate that this emphasis will contribute to the evolution of more accurate quantization techniques in the future, leveraging the versatility of our library to facilitate advancements in the field.

---

### Meta-Review · Area_Chair_GttQ · 2023-12-10

**Metareview:**

The paper received positive feedback from the reviewers. They highlight extensive experiments, motivation, and clarity. At the same time, three of the reviewers suggested that the manuscript's originality and novelty are marginal. Yet, all of the them agreed that the paper is a significant contribution to the community. Hence, the AC decided to accept the manuscript. Congrats!

**Justification For Why Not Higher Score:**

This is a good work, with potential strong impact. Yet, 3 out of 4 reviewers suggested that the novelty is marginal.

**Justification For Why Not Lower Score:**

This is clearly an accept.

---

### Decision · Program_Chairs · 2024-01-16

Accept (poster)